# A CONVOLUTIONAL ENCODER MODEL FOR NEURAL MACHINE TRANSLATION

**Jonas Gehring, Michael Auli, David Grangier, Yann N. Dauphin**
Facebook AI Research

## ABSTRACT

The prevalent approach to neural machine translation relies on bi-directional LSTMs to encode the source sentence. In this paper we present a faster and simpler architecture based on a succession of convolutional layers. This allows to encode the entire source sentence simultaneously compared to recurrent networks for which computation is constrained by temporal dependencies. On WMT'16 English-Romanian translation we achieve competitive accuracy to the state-of-the-art and we outperform several recently published results on the WMT'15 English-German task. Our models obtain almost the same accuracy as a very deep LSTM setup on WMT'14 English-French translation. Our convolutional encoder speeds up CPU decoding by more than two times at the same or higher accuracy as a strong bi-directional LSTM baseline.

## 1 INTRODUCTION

Neural machine translation (NMT) is an end-to-end approach to machine translation (Sutskever et al., 2014). The most successful approach to date encodes the source sentence with a bi-directional recurrent neural network (RNN) into a variable length representation and then generates the translation left-to-right with another RNN where both components interface via a soft-attention mechanism (Bahdanau et al., 2015; Luong et al., 2015a; Bradbury & Socher, 2016; Sennrich et al., 2016b). The recurrent networks are typically parameterized as long short term memory networks (LSTM; Hochreiter et al. 1997) or gated recurrent units (GRU; Cho et al. 2014), often with residual or skip connections (Wu et al., 2016; Zhou et al., 2016) to enable stacking of several layers (§2).

There have been several attempts to use convolutional encoder models for neural machine translation in the past but they were either only applied to rescoring n-best lists of classical systems (Kalchbrenner & Blunsom, 2013) or were not competitive to recurrent alternatives (Cho et al., 2014a). This is despite several attractive properties of convolutional networks. For example, convolutional networks operate over a fixed-size window of the input sequence which enables the simultaneous computation of all features for a source sentence. This contrasts with RNNs which maintain a hidden state of the entire past that prevents parallel computation within a sequence.

Furthermore, a succession of convolutional layers provides a shorter path to capture relationships between elements of a sequence compared to recurrent networks.[1] This also eases learning because the resulting tree-structure applies a fixed number of non-linearities compared to a recurrent neural network. Because processing is bottom-up, all words undergo the same number of transformations, whereas for recurrent networks the first word is over-processed and the last word is transformed only once.

In this paper we show that an architecture based on convolutional layers is very competitive to recurrent encoders. We investigate simple average pooling as well as parameterized convolutions as an alternative for recurrent encoders and enable very deep convolutional encoders by using residual connections (He et al., 2015; §3).

We experiment on several standard datasets and compare our approach to variants of recurrent encoders such as uni-directional and bi-directional LSTMs. On WMT'16 English-Romanian transla-

---

[1] For kernel width $k$ and sequence length $n$ we require $\max\left(1, \left\lceil \frac{n-1}{k-1} \right\rceil\right)$ forwards on a succession of stacked convolutional networks compared to $n$ forwards with a recurrent network.

tion we achieve accuracy that is very competitive to the current state-of-the-art single system result. We perform competitively on WMT'15 English-German, and nearly match the performance of the best WMT'14 English-French system based on a deep LSTM setup when comparing on a commonly used subset of the training data (Zhou et al. 2016; §4, §5).

## 2 RECURRENT NEURAL MACHINE TRANSLATION

The general architecture of the models in this work follows the encoder-decoder approach with soft attention first introduced in Bahdanau et al. (2015). A source sentence $\mathbf{x} = (x_1, \ldots, x_m)$ of $m$ words is processed by an encoder which outputs a sequence of states $\mathbf{z} = (z_1, \ldots, z_m)$.

The decoder is an RNN network that computes a new hidden state $s_{i+1}$ based on the previous state $s_i$, an embedding $g_i$ of the previous target language word $y_i$, as well as a conditional input $c_i$ derived from the encoder output $\mathbf{z}$. We use LSTMs (Hochreiter & Schmidhuber, 1997) for all decoder networks whose state $s_i$ comprises of a cell vector and a hidden vector $h_i$ which is output by the LSTM at each time step. We input $c_i$ into the LSTM by concatenating it to $g_i$.

The translation model computes a distribution over the $V$ possible target words $y_{i+1}$ by transforming the LSTM output $h_i$ via a linear layer with weights $W_o$ and bias $b_o$:

$$p(y_{i+1}|y_1, \ldots, y_i, \mathbf{x}) = \text{softmax}(W_o h_{i+1} + b_o) \in \mathbb{R}^V$$

The conditional input $c_i$ at time $i$ is computed via a simple dot-product style attention mechanism (Luong et al., 2015a). Specifically, we transform the decoder hidden state $h_i$ by a linear layer with weights $W_d$ and $b_d$ to match the size of the embedding of the previous target word $g_i$ and then sum the two representations to yield $d_i$. Conditional input $c_i$ is a weighted sum of attention scores $\mathbf{a_i} \in \mathbb{R}^m$ and encoder outputs $\mathbf{z}$. The attention scores $\mathbf{a_i}$ are determined by a dot product between $h_i$ with each $z_j$, followed by a softmax over the source sequence:

$$d_i = W_d h_i + b_d + g_i \qquad a_{ij} = \frac{\exp\left(d_i^T z_j\right)}{\sum_{t=1}^m \exp\left(d_i^T z_t\right)} \qquad c_i = \sum_{j=1}^m a_{ij} z_j$$

In preliminary experiments, we did not find the MLP attention of Bahdanau et al. (2015) to perform significantly better in terms of BLEU nor perplexity. However, we found the dot-product attention to be more favorable in terms of training and evaluation speed.

We use bi-directional LSTMs to implement recurrent encoders similar to Zhou et al. (2016) which achieved some of the best WMT14 English-French results reported to date. First, each word of the input sequence $\mathbf{x}$ is embedded in distributional space resulting in $\mathbf{e} = (e_1, \ldots, e_m)$. The embeddings are input to two stacks of uni-directional RNNs where the output of each layer is reversed before being fed into the next layer. The first stack takes the original sequence while the second takes the reversed input sequence; the output of the second stack is reversed so that the final outputs of the stacks align. Finally, the top-level hidden states of the two stacks are concatenated and fed into a linear layer to yield $\mathbf{z}$. We denote this encoder architecture as BiLSTM.

## 3 NON-RECURRENT ENCODERS

### 3.1 POOLING ENCODER

A simple baseline for non-recurrent encoders is the pooling model described in Ranzato et al. (2015) which simply averages the embeddings of $k$ consecutive words. Averaging word embeddings does not convey positional information besides that the words in the input are somewhat close to each other. As a remedy, we add position embeddings to encode the absolute position of each source word within a sentence. Each source embedding $e_j$ therefore contains a position embedding $l_j$ as well as the word embedding $w_j$. Position embeddings have also been found helpful in memory networks for question-answering and language modeling (Sukhbaatar et al., 2015). Similar to the recurrent encoder (§2), the attention scores $a_{ij}$ are computed from the pooled representa-

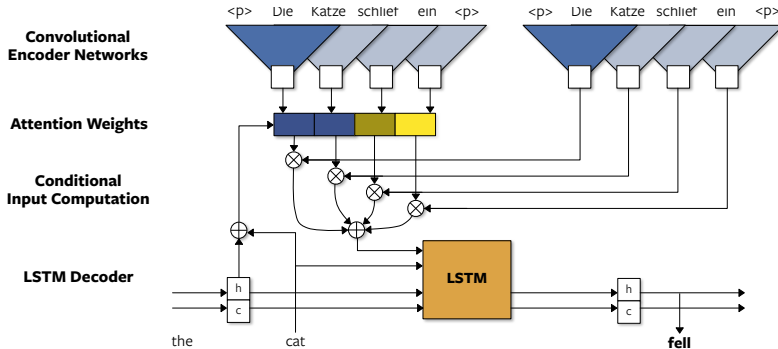

Figure 1: Neural machine translation model with single-layer convolutional encoder networks. CNN-a is on the left and CNN-c is at the right. Embedding layers are not shown.

tions $z_j$, however, the conditional input $c_i$ is a weighted sum of the embeddings $e_j$, not $z_j$, i.e.,

$$e_j = w_j + l_j \qquad\qquad z_j = \frac{1}{k}\sum_{t=-\lfloor k/2\rfloor}^{\lfloor k/2\rfloor} e_{j+t} \qquad\qquad c_i = \sum_{j=1}^{m} a_{ij}e_j$$

The input sequence is padded prior to pooling such that the encoder output matches the input length $|\mathbf{z}| = |\mathbf{x}|$. We set $k$ to 5 in all experiments as Ranzato et al. (2015).

## 3.2 CONVOLUTIONAL ENCODER

A straightforward extension of pooling is to learn the kernel in a convolutional neural network (CNN). The encoder output $z_j$ contains information about a fixed-sized context depending on the kernel width $k$ but the desired context width may vary. This can be addressed by stacking several layers of convolutions followed by non-linearities: additional layers increase the total context size while non-linearities can modulate the effective size of the context as needed. For instance, stacking 5 convolutions with kernel width $k = 3$ results in an input field of 11 words, i.e., each output depends on 11 input words, and the non-linearities allow the encoder to exploit the full input field, or to concentrate on fewer words as needed.

To ease learning for deep encoders, we add residual connections from the input of each convolution to the output and then apply the non-linear activation function to the output (tanh; He et al., 2015); the non-linearities are therefore not 'bypassed'. Multi-layer CNNs are constructed by stacking several blocks on top of each other. The CNNs do not contain pooling layers which are commonly used for down-sampling, i.e., the full source sequence length will be retained after the network has been applied. Similar to the pooling model, the convolutional encoder uses position embeddings.

The final encoder consists of two stacked convolutional networks (Figure 1): CNN-a produces the encoder output $z_j$ to compute the attention scores $\mathbf{a_i}$, while the conditional input $c_i$ to the decoder is computed by summing the outputs of CNN-c,

$$z_j = \text{CNN-a}(\mathbf{e})_j \qquad\qquad c_i = \sum_{j=1}^{m} a_{ij}\,\text{CNN-c}(\mathbf{e})_j.$$

In practice, we found that two different CNNs resulted in better perplexity as well as BLEU compared to using a single one (§5.3). We also found this to perform better than directly summing the $e_i$ without transformation as for the pooling model.

## 3.3 RELATED WORK

There are several past attempts to use convolutional encoders for neural machine translation, however, to our knowledge none of them were able to match the performance of recurrent encoders. Kalchbrenner & Blunsom (2013) introduce a convolutional sentence encoder in which a multi-layer CNN generates a fixed sized embedding for a source sentence, or an n-gram representation followed by transposed convolutions for directly generating a per-token decoder input. The latter requires

the length of the translation prior to generation and both models were evaluated by rescoring the output of an existing translation system. Cho et al. (2014a) propose a gated recursive CNN which is repeatedly applied until a fixed-size representation is obtained but the recurrent encoder achieves higher accuracy. In follow-up work, the authors improved the model via a soft-attention mechanism but did not re-consider convolutional encoder models (Bahdanau et al., 2015).

Concurrently to our work, Kalchbrenner et al. (2016) have introduced convolutional translation models without an explicit attention mechanism but their approach does not yet result in state-of-the-art accuracy. Lamb & Xie (2016) also proposed a multi-layer CNN to generate a fixed-size encoder representation but their work lacks quantitative evaluation in terms of BLEU. Meng et al. (2015) and Tu et al. (2015) applied convolutional models to score phrase-pairs of traditional phrase-based and dependency-based translation models. Convolutional architectures have also been successful in language modeling but so far failed to outperform LSTMs (Pham et al., 2016).

# 4 EXPERIMENTAL SETUP

## 4.1 DATASETS

We evaluate different encoders and ablate architectural choices on a small dataset from the German-English machine translation track of IWSLT 2014 (Cettolo et al., 2014) with a similar setting to Ranzato et al. (2015). Unless otherwise stated, we restrict training sentences to have no more than 175 words; test sentences are not filtered. This is a higher threshold compared to other publications but ensures proper training of the position embeddings for non-recurrent encoders; the length threshold did not significantly effect recurrent encoders. Length filtering results in 167K sentence pairs and we test on the concatenation of *tst2010, tst2011, tst2012, tst2013* and *dev2010* comprising 6948 sentence pairs.[2] Our final results are on three major WMT tasks:

**WMT'16 English-Romanian.** We use the same data and pre-processing as Sennrich et al. (2016b) and train on 2.8M sentence pairs.[3] Our model is word-based instead of relying on byte-pair encoding (Sennrich et al., 2016a). We evaluate on *newstest2016*.

**WMT'15 English-German.** We use all available parallel training data, namely Europarl v7, Common Crawl and News Commentary v10 and apply the standard Moses tokenization to obtain 3.9M sentence pairs (Koehn et al., 2007). We report results on *newstest2015*.

**WMT'14 English-French.** We use a commonly used subset of 12M sentence pairs (Schwenk, 2014), and remove sentences longer than 150 words. This results in 10.7M sentence-pairs for training. Results are reported on *ntst14*.

A small subset of the training data serves as validation set (5% for IWSLT'14 and 1% for WMT) for early stopping and learning rate annealing (§4.3). For IWSLT'14, we replace words that occur fewer than 3 times with a `<unk>` symbol, which results in a vocabulary of 24158 English and 35882 German word types. For WMT datasets, we retain 200K source and 80K target words. For English-French only, we set the target vocabulary to 30K types to be comparable with previous work.

## 4.2 MODEL PARAMETERS

We use 512 hidden units for both recurrent encoders and decoders. We reset the decoder hidden states to zero between sentences. For the convolutional encoder, 512 hidden units are used for each layer in CNN-a, while layers in CNN-c contain 256 units each. All embeddings, including the output produced by the decoder before the final linear layer, are of 256 dimensions. On the WMT corpora, we find that we can improve the performance of the bi-directional LSTM models (BiLSTM) by using 512-dimensional word embeddings.

Model weights are initialized from a uniform distribution within $[-0.05, 0.05]$. For convolutional layers, we use a uniform distribution of $\left[-kd^{-0.5}, kd^{-0.5}\right]$, where $k$ is the kernel width (we use 3

---

[2]Different to the other datasets, we lowercase the training data and evaluation with case-insensitive BLEU.

[3]We followed the pre-processing of `https://github.com/rsennrich/wmt16-scripts/blob/master/sample/preprocess.sh` and added the back-translated data from `http://data.statmt.org/rsennrich/wmt16_backtranslations/en-ro`.

throughout this work) and $d$ is the input size for the first layer and the number of hidden units for subsequent layers (Collobert et al., 2011b). For CNN-c, we transform the input and output with a linear layer each to match the smaller embedding size. The model parameters were tuned on IWSLT'14 and cross-validated on the larger WMT corpora.

## 4.3 OPTIMIZATION

Recurrent models are trained with Adam as we found them to benefit from aggressive optimization. We use a step width of $3.125 \cdot 10^{-4}$ and early stopping based on validation perplexity (Kingma & Ba, 2014). For non-recurrent encoders, we obtain best results with stochastic gradient descent (SGD) and annealing: we use a learning rate of 0.1 and once the validation perplexity stops improving, we reduce the learning rate by an order of magnitude each epoch until it falls below $10^{-4}$.

For all models, we use mini-batches of 32 sentences for IWSLT'14 and 64 for WMT. We use truncated back-propagation through time to limit the length of target sequences per mini-batch to 25 words. Gradients are normalized by the mini-batch size. We re-normalize the gradients if their norm exceeds 25 (Pascanu et al., 2013). Gradients of convolutional layers are scaled by $\text{sqrt}(\dim(input))^{-1}$ similar to Collobert et al. (2011b). We use dropout on the embeddings and decoder outputs $h_i$ with a rate of 0.2 for IWSLT'14 and 0.1 for WMT (Srivastava et al., 2014). All models are implemented in Torch (Collobert et al., 2011a) and trained on a single GPU.

## 4.4 EVALUATION

We report accuracy of single systems by training several identical models with different random seeds (5 for IWSLT'14, 3 for WMT) and pick the one with the best validation perplexity for final BLEU evaluation. Translations are generated by a beam search and we normalize log-likelihood scores by sentence length. On IWSLT'14 we use a beam width of 10 and for WMT models we tune beam width and word penalty on a separate test set, that is *newsdev2016* for WMT'16 English-Romanian, *newstest2014* for WMT'15 English-German and *ntst1213* for WMT'14 English-French.[4] The word penalty adds a constant factor to log-likelihoods, except for the end-of-sentence token.

Prior to scoring the generated translations against the respective references, we perform unknown word replacement based on attention scores (Jean et al., 2015). Unknown words are replaced by looking up the source word with the maximum attention score in a pre-computed dictionary. If the dictionary contains no translation, then we simply copy the source word. Dictionaries were extracted from the aligned training data that was aligned with `fast_align` (Dyer et al., 2013). Each source word is mapped to the target word it is most frequently aligned to.

For convolutional encoders with stacked CNN-c layers we noticed for some models that the attention maxima were consistently shifted by one word. We determine this per-model offset on the above-mentioned development sets and correct for it. Finally, we compute case-sensitive tokenized BLEU, except for WMT'16 English-Romanian where we use *detokenized* BLEU to be comparable with Sennrich et al. (2016b).[5]

## 5 RESULTS

### 5.1 RECURRENT VS. NON-RECURRENT ENCODERS

We first compare recurrent and non-recurrent encoders in terms of perplexity and BLEU on IWSLT'14 with and without position embeddings (§3.1) and include a phrase-based system (Koehn et al., 2007). Table 1 shows that a single-layer convolutional model with position embeddings (Convolutional) can outperform both a uni-directional LSTM encoder (LSTM) as well as a bi-directional LSTM encoder (BiLSTM). Next, we increase the depth of the convolutional encoder. We choose a good setting by independently varying the number of layers in CNN-a and CNN-c between 1 and 10 and obtained best validation set perplexity with six layers for CNN-a and three layers for CNN-c.

---

[4]Specifically, we select a beam from $\{5, 10\}$ and a word penalty from $\{0, -0.5, -1, -1.5\}$

[5]`https://github.com/moses-smt/mosesdecoder/blob/617e8c8ed1630fb1d1/scripts/generic/{multi-bleu.perl,mteval-v13a.pl}`

| System/Encoder | BLEU words + pos | BLEU words | PPL words + pos |
|---|---|---|---|
| Phrase-based | – | 28.4 | – |
| LSTM | 27.4 | 27.3 | 10.8 |
| BiLSTM | 29.7 | 29.8 | 9.9 |
| Pooling | 26.1 | 19.7 | 11.0 |
| Convolutional | 29.9 | 20.1 | 9.1 |
| Deep Convolutional 6/3 | 30.4 | 25.2 | 8.9 |

Table 1: Accuracy of encoders with position features (words + pos) and without (words) in terms of BLEU and perplexity (PPL) on IWSLT'14 German to English translation; results include unknown word replacement. Deep Convolutional 6/3 is the only multi-layer configuration, more layers for the LSTMs did not improve accuracy on this dataset.

This configuration outperforms BiLSTM by 0.7 BLEU (Deep Convolutional 6/3). We investigate depth in the convolutional encoder more in §5.3.

Among recurrent encoders, the BiLSTM is 2.3 BLEU better than the uni-directional version. The simple pooling encoder which does not contain any parameters is only 1.3 BLEU lower than a uni-directional LSTM encoder and 3.6 BLEU lower than BiLSTM. The results without position embeddings (words) show that position information is crucial for convolutional encoders. In particular for shallow models (Pooling and Convolutional), whereas deeper models are less effected. Recurrent encoders do not benefit from explicit position information because this information can be naturally extracted through the sequential computation.

When tuning model settings, we generally observe good correlation between perplexity and BLEU. However, for convolutional encoders perplexity gains translate to smaller BLEU improvements compared to recurrent counterparts (Table 1). We observe a similar trend on larger datasets.

## 5.2 EVALUATION ON WMT CORPORA

Next, we evaluate the BiLSTM encoder and the convolutional encoder architecture on three larger tasks and compare against previously published results. On WMT'16 English-Romanian translation we compare to Sennrich et al. (2016b), the winning single system entry for this language pair. Their model consists of a bi-directional GRU encoder, a GRU decoder and MLP-based attention. They use byte pair encoding (BPE) to achieve open-vocabulary translation and dropout in all components of the neural network to achieve 28.1 BLEU; we use the same pre-processing but no BPE (§4).

The results (Table 2) show that a deep convolutional encoder performs competitively to the state of the art on this dataset (Sennrich et al., 2016b). Our bi-directional LSTM encoder baseline is 0.7 BLEU lower than Sennrich et al. (2016b) but uses only 512 hidden units compared to 1024. A single-layer convolutional encoder with embedding size 256 performs very competitively at 27.1 BLEU which is only 0.3 BLEU below the BiLSTM baseline. Increasing the number of convolutional layers to 8 in CNN-a and 4 in CNN-c achieves 27.8 BLEU which performs well above this baseline.

On WMT'15 English to German, we compare to a BiLSTM baseline and prior work: Jean et al. (2015) introduce a large output vocabulary; the decoder of Chung et al. (2016) operates on the character-level; Yang et al. (2016) uses LSTMs instead of GRUs and feeds the conditional input to the output layer as well as to the decoder.

Our single-layer BiLSTM baseline performs competitively compared to prior work and a two-layer BiLSTM performs about 0.4 BLEU better at 23.6 BLEU. Previous work also used multi-layer setups, e.g., Chung et al. (2016) has two layers both in the encoder and the decoder with 1024 hidden units, and Yang et al. (2016) use 1000 hidden units per LSTM. We use 512 hidden units for both LSTM and convolutional encoders. A single-layer CNN encoder (Convolutional) achieves 22.0 BLEU which is significantly lower than the two-layer BiLSTM. However, adding additional layers (Deep Convolutional 8/4) achieves the same accuracy as the two-layer BiLSTM and a 15 layer CNN-a outperforms it by 0.7 BLEU (Deep Convolutional 15/5). The latter performs competitively to the best published results which use decoder improvements that may benefit our setup as well.

| WMT'16 English-Romanian | Encoder | Vocabulary | BLEU |
|---|---|---|---|
| Sennrich et al. (2016b) | BiGRU | BPE 90K | 28.1 |
| Single-layer decoder | BiLSTM | 80K | 27.4 |
| | Convolutional | 80K | 27.1 |
| | Deep Convolutional 8/4 | 80K | 27.8 |

| WMT'15 English-German | Encoder | Vocabulary | BLEU |
|---|---|---|---|
| Jean et al. (2015) RNNsearch-LV | BiGRU | 500K | 22.4 |
| Chung et al. (2016) BPE-Char | BiGRU | Char 500 | 23.9 |
| Yang et al. (2016) RNNSearch + UNK replace | BiLSTM | 50K | 24.3 |
| + *recurrent attention* | BiLSTM | 50K | 25.0 |
| Single-layer decoder | BiLSTM | 80K | 23.2 |
| | 2-layer BiLSTM | 80K | 23.6 |
| | Convolutional | 80K | 22.0 |
| | Deep Convolutional 8/4 | 80K | 23.6 |
| | Deep Convolutional 15/5 | 80K | 24.3 |

| WMT'14 English-French (12M) | Encoder | Vocabulary | BLEU |
|---|---|---|---|
| Bahdanau et al. (2015) RNNsearch | BiGRU | 30K | 28.5 |
| Luong et al. (2015b) Single LSTM | 6-layer LSTM | 40K | 32.7 |
| Jean et al. (2014) RNNsearch-LV | BiGRU | 500K | 34.6 |
| Zhou et al. (2016) Deep-Att | Deep BiLSTM | 30K | 35.9 |
| Single-layer decoder | BiLSTM | 30K | 34.6 |
| | Deep Convolutional 8/4 | 30K | 34.6 |
| Two-layer decoder | 2-layer BiLSTM | 30K | 35.3 |
| | Deep Convolutional 20/5 | 30K | 35.7 |

Table 2: Accuracy on three WMT tasks, including results published in previous work. For deep convolutional encoders, we include the number of layers in CNN-a and CNN-c, respectively.

Finally, we evaluate on the larger WMT'14 English-French corpus. On this dataset the recurrent architectures benefit from an additional layer both in the encoder and the decoder. For a single-layer decoder, a deep convolutional encoder matches the BiLSTM accuracy and for a two-layer decoder, our very deep convolutional encoder with up to 20 layers outperforms the BiLSTM by 0.4 BLEU. It has 40% fewer parameters than the BiLSTM due to the smaller embedding sizes. We also outperform several previous systems, including the very deep encoder-decoder model proposed by Luong et al. (2015a). Our best result is just 0.2 BLEU below Zhou et al. (2016) who use a very deep LSTM setup with a 9-layer encoder, a 7-layer decoder, shortcut connections and extensive regularization with dropout and L2 regularization.

## 5.3 CONVOLUTIONAL ENCODER ARCHITECTURE DETAILS

We next motivate our design of the convolutional encoder (§3.2). We use the smaller IWSLT'14 German-English setup without unknown word replacement to enable fast experimental turn-around. BLEU results are averaged over three training runs initialized with different seeds.

Figure 2 shows accuracy for a different number of layers of both CNNs with and without residual connections. Our first observation is that computing the conditional input $c_i$ directly over embeddings $e$ (line "without CNN-c") is already working well at 28.3 BLEU with a single CNN-a layer and at 29.1 BLEU for CNN-a with 7 layers (Figure 2a). Increasing the number of CNN-c layers is beneficial up to three layers and beyond this we did not observe further improvements. Similarly, increasing the number of layers in CNN-a beyond six does not increase accuracy on this relatively small dataset. In general, choosing two to three times as many layers in CNN-a as in CNN-c is a good rule of thumb. Without residual connections, the model fails to utilize the increase in modeling power from additional layers, and performance drops significantly for deeper encoders (Figure 2b).

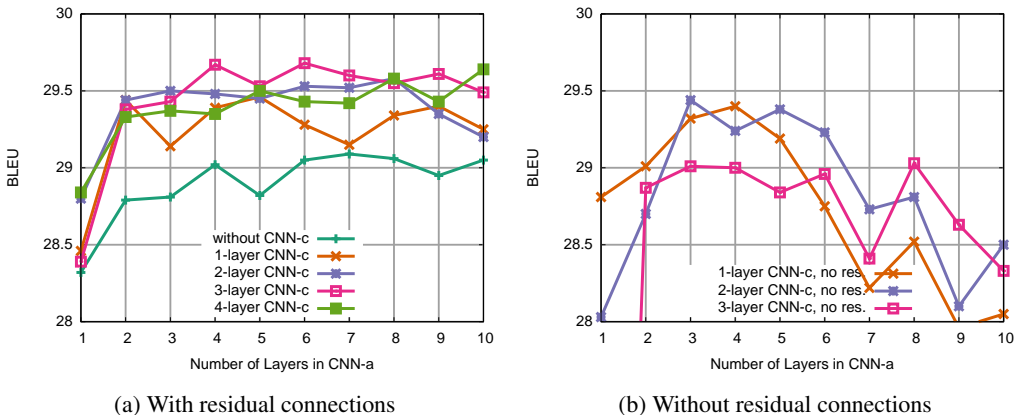

(a) With residual connections  (b) Without residual connections

Figure 2: Effect of encoder depth on IWSLT'14 with and without residual connections. The x-axis varies the number of layers in CNN-a and curves show different CNN-c settings.

| Encoder | Words/s | BLEU |
|---|---|---|
| BiLSTM | 139.7 | 22.4 |
| Deep Conv. 6/3 | 187.9 | 23.1 |

(a) IWSLT'14 German-English generation speed on *tst2013* with beam size 10.

| Encoder | Words/s | BLEU |
|---|---|---|
| 2-layer BiLSTM | 109.9 | 23.6 |
| Deep Conv. 8/4 | 231.1 | 23.7 |
| Deep Conv. 15/5 | 203.3 | 24.0 |

(b) WMT'15 English-German generation speed on *newstest2015* with beam size 5.

Table 3: Generation speed in source words per second on a single CPU core.

Our convolutional architecture relies on two sets of networks, CNN-a for attention score computation $a_i$ and CNN-c for the conditional input $c_i$ to be fed to the decoder. We found that using the same network for both tasks, similar to recurrent encoders, resulted in poor accuracy of 22.9 BLEU. This compares to 28.5 BLEU for separate single-layer networks, or 28.3 BLEU when aggregating embeddings for $c_i$. Increasing the number of layers in the single network setup did not help. Figure 2(a) suggests that the attention weights (CNN-a) need to integrate information from a wide context which can be done with a deep stack. At the same time, the vectors which are averaged (CNN-c) seem to benefit from a shallower, more local representation closer to the input words. Two stacks are an easy way to achieve these contradicting requirements.

In Appendix A we visualize attention scores and find that alignments for CNN encoders are less sharp compared to BiLSTMs, however, this does not affect the effectiveness of unknown word replacement once we adjust for shifted maxima. In Appendix B we investigate whether deep convolutional encoders are required for translating long sentences and observe that even relatively shallow encoders perform well on long sentences.

## 5.4 TRAINING AND GENERATION SPEED

For training, we use the fast CuDNN LSTM implementation for layers without attention and experiment on IWSLT'14 with batch size 32. The single-layer BiLSTM model trains at 4300 target words/second, while as the 6/3 deep convolutional encoder compares at 5500 words/second on an NVidia Tesla M40 GPU. We do not observe shorter overall training time since SGD converges slower than Adam which we use for BiLSTM models.

We measure generation speed on an Intel Haswell CPU clocked at 2.50GHz with a single thread for BLAS operations. We use vocabulary selection which can speed up generation by up to a factor of ten at no cost in accuracy via making the time to compute the final output layer negligible (Mi et al., 2016; L'Hostis et al., 2016). This shifts the focus from the efficiency of the encoder to the efficiency of the decoder. On IWSLT'14 (Table 3a) the convolutional encoder increases the speed of

the overall model by a factor of 1.35 compared to the BiLSTM encoder while improving accuracy by 0.7 BLEU. In this setup both encoders models have the same hidden layer and embedding sizes.

On the larger WMT'15 English-German task (Table 3b) the convolutional encoder speeds up generation by 2.1 times compared to a two-layer BiLSTM. This corresponds to 231 source words/second with beam size 5. Our best model on this dataset generates 203 words/second but at slightly lower accuracy compared to the full vocabulary setting in Table 2. The recurrent encoder uses larger embeddings than the convolutional encoder which were required for the models to match in accuracy.

The smaller embedding size is not the only reason for the speed-up. In Table 3a (a), we compare a Conv 6/3 encoder and a BiLSTM with equal embedding sizes. The convolutional encoder is still 1.34x faster (at 0.7 higher BLEU) although it requires roughly 1.6x as many FLOPs. We believe that this is likely due to better cache locality for convolutional layers on CPUs: an LSTM with fused gates[6] requires two big matrix multiplications with different weights as well as additions, multiplications and non-linearities for each source word, while the output of each convolutional layer can be computed as whole with a single matrix multiply.

For comparison, the quantized deep LSTM-based model in Wu et al. (2016) processes 106.4 words/second for English-French on a CPU with 88 cores and 358.8 words/second on a custom TPU chip. The optimized RNNsearch model and C++ decoder described by Junczys-Dowmunt et al. (2016) translates 265.3 words/s on a CPU with a similar vocabulary selection technique, computing 16 sentences in parallel, i.e., 16.6 words/s on a single core.

## 6 CONCLUSION

We introduced a simple encoder model for neural machine translation based on convolutional networks. This approach is more parallelizable than recurrent networks and provides a shorter path to capture long-range dependencies in the source. We find it essential to use source position embeddings as well as different CNNs for attention score computation and conditional input aggregation.

Our experiments show that convolutional encoders perform on par or better than baselines based on bi-directional LSTM encoders. In comparison to other recent work, our deep convolutional encoder is very competitive to the best published results to date (WMT'16 English-Romanian) which are obtained with significantly more complex models (WMT'14 English-French) or stem from improvements that are orthogonal to our work (WMT'15 English-German). Our architecture also leads to large generation speed improvements: translation models with our convolutional encoder can translate twice as fast as strong baselines with bi-directional recurrent encoders.

Future work includes better training to enable faster convergence with the convolutional encoder to better leverage the higher processing speed. Our fast architecture is interesting for character level encoders where the input is significantly longer than for words. Also, we plan to investigate the effectiveness of our architecture on other sequence-to-sequence tasks, e.g. summarization, constituency parsing, dialog modeling.

## ACKNOWLEDGMENTS

We would like to thank Sumit Chopra and Marc'Aurelio Ranzato for helpful discussions related to this work.

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

## A ALIGNMENT VISUALIZATION

In Figure 4 and Figure 5, we plot attention scores for a sample WMT'15 English-German and WMT'14 English-French translation with BiLSTM and deep convolutional encoders. The translation is on the x-axis and the source sentence on the y-axis.

The attention scores of the BiLSTM output are sharp but do not necessarily represent a correct alignment. For CNN encoders the scores are less focused but still indicate an approximate source location, e.g., in Figure 4b, when moving the clause "over 1,000 people were taken hostage" to the back of the translation. For some models, attention maxima are consistently shifted by one token as both in Figure 4b and Figure 5b.

Interestingly, convolutional encoders tend to focus on the last token (Figure 4b) or both the first and last tokens (Figure 5b). Motivated by the hypothesis that the this may be due to the decoder depending on the length of the source sentence (which it cannot determine without position embeddings), we explicitly provided a distributed representation of the input length to the decoder and attention module. However, this did not cause a change in attention patterns nor did it improve translation accuracy.

## B PERFORMANCE BY SENTENCE LENGTH

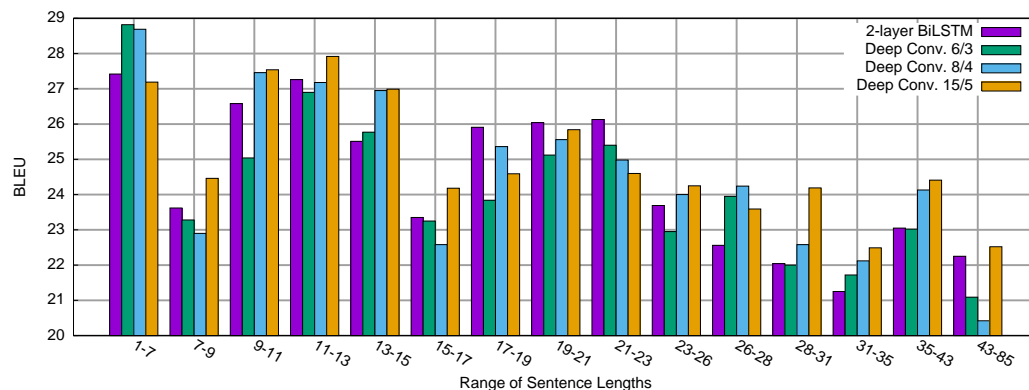

Figure 3: BLEU per sentence length on WMT'15 English-German *newstest2015*. The test set is partitioned into 15 equally-sized buckets according to source sentence length.

One characteristic of our convolutional encoder architecture is that the context over which outputs are computed depends on the number of layers. With bi-directional RNNs, every encoder output depends on the entire source sentence. In Figure 3, we evaluate whether limited context affects the translation quality on longer sentences of WMT'15 English-German which often requires moving verbs over long distances. We sort the *newstest2015* test set by source length, partition it into 15 equally-sized buckets, and compare the BLEU scores of models listed in Table 2 on a per-bucket basis.

There is no clear evidence for sub-par translations on sentences that are longer than the observable context per encoder output. We include a small encoder with a 6-layer CNN-c and a 3-layer CNN-a in the comparison which performs worse than a 2-layer BiLSTM (23.3 BLEU vs 23.6). With 6 convolutional layers at kernel width 3, each encoder output contains information of 13 adjacent source words . Looking at the accuracy for sentences with 15 words or more, this relatively shallow CNN is either on par or better than the BiLSTM for 5 out of 10 buckets; the BiLSTM has access to the entire source context. Similar observations can be made for the deeper convolutional encoders.

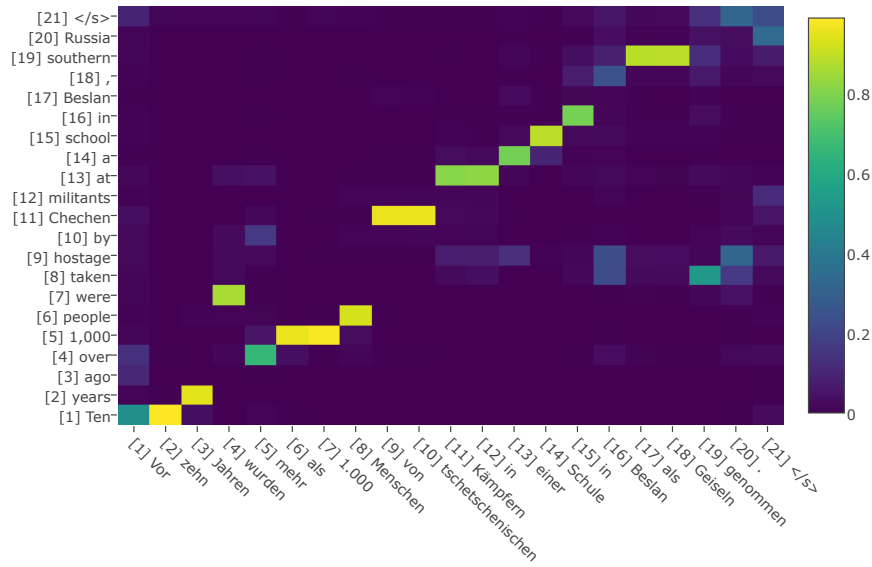

(a) 2-layer BiLSTM encoder.

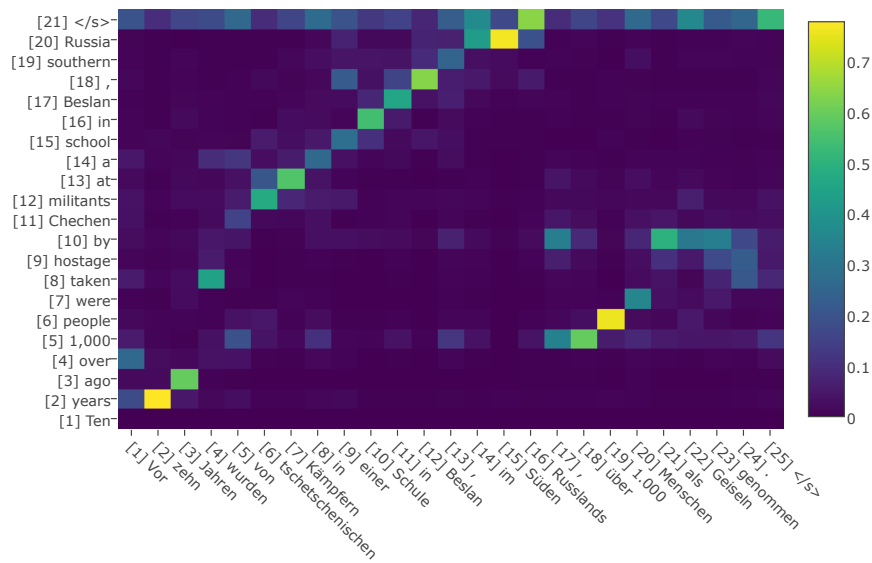

(b) Deep convolutional encoder with 15-layer CNN-a and 5-layer CNN-c.

Figure 4: Attention scores for WMT'15 English-German translation for a sentence of *newstest2015*.

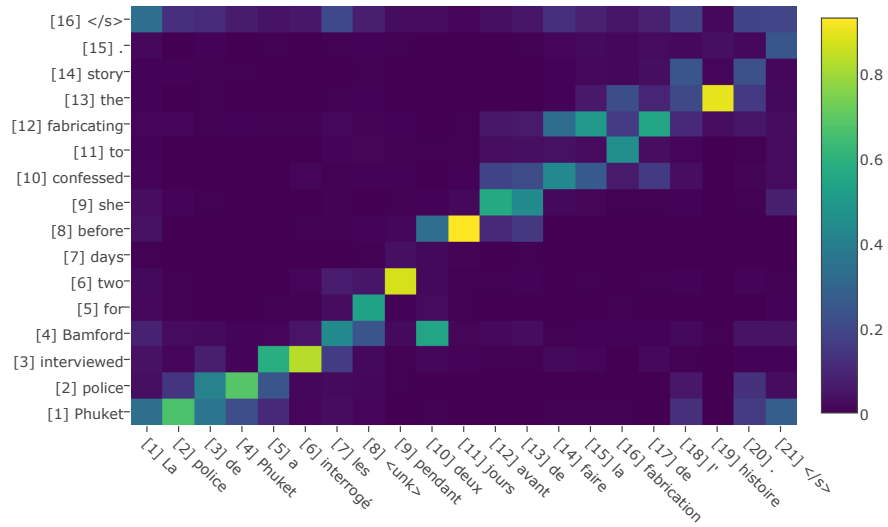

(a) 2-layer BiLSTM encoder.

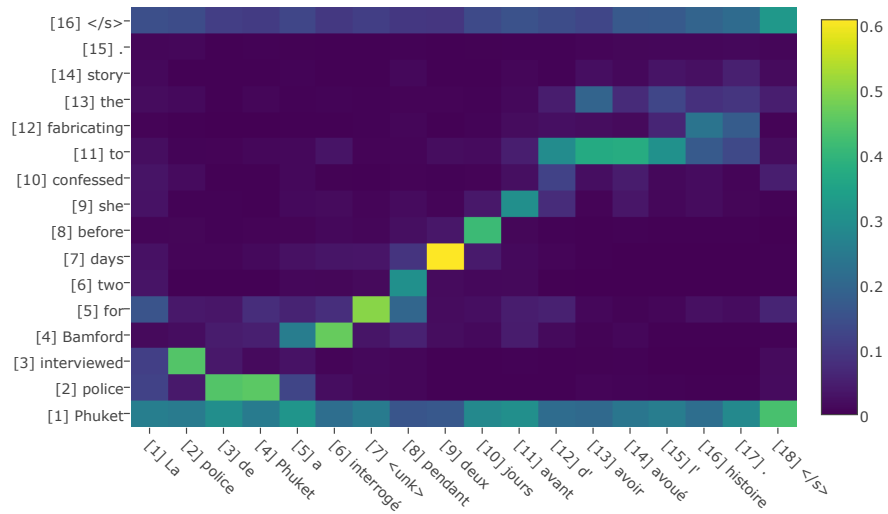

(b) Deep convolutional encoder with 20-layer CNN-a and 5-layer CNN-c.

Figure 5: Attention scores for WMT'14 English-French translation for a sentence of *ntst14*.

