# Peer review of "A Convolutional Encoder Model for Neural Machine Translation"

_ICLR 2017 — rejected_

[Public Comment · Rico Sennrich · 16 Nov 2016]
**multi-bleu.perl**

I want to draw attention to the fact that you compare tokenized BLEU results (with multi-bleu.perl) and detokenized BLEU results (with mteval-v13a.pl). The two should *never* be mixed in the same table, as the tokenization will have a big effect on results. Even when comparing systems that all have tokenized BLEU, and all use the Moses tokenizer, using different parameters (such as the "-a" option for aggressive hyphen splitting) will skew the results.

Detokenized BLEU is standard for WMT, and reported by Sennrich et al. (2016a,b).

I re-ran BLEU on our EN-RO system for comparison:

detokenized BLEU, mteval-v13a.pl: 28.1 BLEU
tokenized BLEU, multi-bleu.perl: 29.4 BLEU

your reported result (multi-bleu.perl): 28.5

[Public Comment · Ozan Caglayan · 17 Nov 2016]
**Small typo**

Hi,

The conditional input is denoted by both c_i and c_t in sections 2 and 3.1 respectively. c_i is reused in section 3.2

[Official Review · AnonReviewer1 · rating 6 · confidence 4 · 16 Dec 2016]
**Good paper, but very incremental**

The paper reports a very clear and easy to understand result that a convolutional network can be used instead of the recurrent encoder for neural machine translation. 

Apart from the known architectural elements, such as convolution, pooling, residual connections, position embeddings, the paper features one unexpected architectural twist: two stacks of convolutions, one for computing alignment and another for computing the representations.
The empirical evidence that this was necessary is provided, however the question of *why* it is necessary remains open. 

The experimental evaluation is very extensive and leaves no doubt that the proposed approach works well. The convnet-based model was faster at evaluation, but it is not very clear what is the main speed-up factor. It’s however hard to argue against the fact that the speed advantage of convnets is likely to increase if a more parallel implementation is considered. 

My main concern is whether or not the paper is appropriate for ICLR, because the contribution is quite incremental and rather application-specific. ACL, EMNLP and other NLP conferences would be a better venue, I think.

[Official Review · AnonReviewer3 · rating 6 · confidence 5 · 17 Dec 2016]
**A well-executed NLP paper**

This paper is the first (I believe) to establish a simple yet important result that Convnets for NMT encoders can be competitive to RNNs. The authors present a convincing set of results over many translation tasks and compare with very competitive baselines. I also appreciate the detailed report on training and generation speed. I find it's very interesting when position embeddings turn out to be hugely important (beside residual connections); unfortunately, there is little analysis to shed more lights on this aspect and perhaps compare other ways of capturing positions (a wild guess might be to use embeddings that represent some form of relative positions). The only concern I have (similar to the other reviewer) is that this paper perhaps fits better in an NLP conference.

One minor comment: it's slight strange that this well-executed paper doesn't have a single figure on the proposed architecture :) It will also be even better to draw a figure for the biLSTM architecture as well (it does take some effort to understand the last paragraph in Section 2, especially the part on having a linear layer to compute z).

[Official Review · AnonReviewer4 · rating 7 · confidence 3 · 19 Dec 2016 (modified: 25 Jan 2017)]
**Final Review: good paper applying CNN to translation to match bi-LSTM baseline**

The system described works comparably to bi-directional LSTM baseline for NMT, and CNN's are naturally parallelizable.

Key ideas include the use of two stacked CNN's (one for each of encoding and decoding) for translation, with res connections and position embeddings. The use of CNN's for translation has been attempted previously (as described by the authors), but presumably it is the authors' combination of various architectural choices (attention, position embeddings, etc) that make the present system competitive with RNN's, whereas earlier attempts were not. They describe system's sensitivity to some of these choices (e.g. experiments to choose appropriate number of layers in each of the CNN's).

The experimental results are well reported in detail.

One or two figures would definitely be required to help clarify the architecture.

This paper is less about new ways of learning representations than about the combination of choices made (over the set of existing techniques) in order to get the good results that they do on the reported NMT tasks. In this respect, while I am fairly confident that the paper represents good work in machine learning, I am not quite as confident about its fit for this particular conference.

[Public Comment · ICLR 2017 conference · 07 Jan 2017]
**Authors: Please post a rebuttal**

Authors, It would be great to have a rebuttal for this paper as reviewers will be discussing over the next week. Thanks.

[Final Decision · Program Chairs · 06 Feb 2017]
**ICLR committee final decision**

This work demonstrates architectural choices to make conv nets work for NMT. In general the reviewers liked the work and were convinced by the results but found the main contributions to be "incremental". 
 
 Pros:
 - Clarity: The work was clearly presented, and besides for minor comments (diagrams) the reviewers understood the work
 - Quality: The experimental results were thorough, "very extensive and leaves no doubt that the proposed approach works well".
 
 Mixed:
 - Novelty: There is appreciation that the work is novel. However as the work is somewhat "application-specific" the reviewers felt the technical contribution was not an overwhelming contribution.
 - Impact: While some speed ups were shown, not all reviewers were convinced that the benefit was sufficient, or "main speed-up factor(s)" were. 
 
 This work is clearly worthwhile, but the reviews place it slightly below the top papers in this area.